# A binary system in the S cluster close to the supermassive black hole Sagittarius A*

Florian Peißker [1] ✉, Michal Zajaček [1,2], Lucas Labadie [1], Emma Bordier [1], Andreas Eckart [1,3], Maria Melamed[1] & Vladimír Karas [4]

High-velocity stars and peculiar G objects orbit the central supermassive black hole (SMBH) Sagittarius A* (Sgr A*). Together, the G objects and high-velocity stars constitute the S cluster. In contrast with theoretical predictions, no binary system near Sgr A* has been identified. Here, we report the detection of a spectroscopic binary system in the S cluster with the masses of the components of $2.80 \pm 0.50\,M_\odot$ and $0.73 \pm 0.14\,M_\odot$, assuming an edge-on configuration. Based on periodic changes in the radial velocity, we find an orbital period of $372\pm3$ days for the two components. The binary system is stable against the disruption by Sgr A* due to the semi-major axis of the secondary being $1.59\pm0.01$ AU, which is well below its tidal disruption radius of approximately 42.4 AU. The system, known as D9, shows similarities to the G objects. We estimate an age for D9 of $2.7^{+1.9}_{-0.3} \times 10^6$ yr that is comparable to the timescale of the SMBH-induced von Zeipel-Lidov-Kozai cycle period of about $10^6$ yr, causing the system to merge in the near future. Consequently, the population of G objects may consist of pre-merger binaries and post-merger products. The detection of D9 implies that binary systems in the S cluster have the potential to reside in the vicinity of the supermassive black hole Sgr A* for approximately $10^6$ years.

The central parsec around the supermassive black hole (SMBH) Sgr A* contains a large number of stars that constitute the Nuclear Star Cluster (NSC)[1], which is one of the densest and most massive stellar systems in the Galaxy. These stars vary in terms of their ages, masses, sizes, and luminosities[2]. In the vicinity of Sgr A* of about 40 mpc, there is a high concentration of stars[3] that orbit the black hole at velocities of up to several thousand km/s[4,5] inside the S cluster. The presence of stars close the Sgr A* is not surprising because it was expected that old and evolved stars would gradually descend towards Sgr A* due to the cluster relaxation timescale of ~$10^{10}$ yr[6]. This is because star formation is significantly inhibited by tidal forces and high energetic radiation in the vicinity of the SMBH. In fact, a cusp of late-type stars with stellar ages of >$3 \times 10^9$ yr was identified[7]. Interestingly, these late-type stars coexist with massive early-type S cluster members that exhibit an average age of ~$4$–$6 \times 10^6$ yr[8,9], resulting in the formulation of the

"paradox of youth"[10]. Until now, no companions have been identified for these young B-type stars[11], although binary rates close to 100% have been proposed[12]. Therefore, the presence of binary systems in the S cluster is a crucial question to investigate the dynamical evolution of stars in the vicinity of Sgr A*[13,14]. Given that the evolution of high-mass stars is altered by their binary interactions[15], it is important to understand the prevalence of putative binary systems in this cluster.

In this work, we present the detection of a spectroscopic binary in the S cluster. Based on the photometric characteristics of the binary system, known as D9, it can be considered to be a member of the G-object population[16,17]. The age of the system is ~$2.7 \times 10^6$yr, which is comparable to the von Zeipel-Lidov-Kozai cycle period of approximately $10^6$ years. The dusty source D9 is most likely composed of a Herbig Ae/Be star associated with the primary. The lower-mass companion can be classified as a T-Tauri star. In the near future, the binary

[1].Physikalisches Institut, Universität zu Köln, Zülpicher Str. 77, Cologne 50937, Germany. [2]Department of Theoretical Physics and Astrophysics, Masaryk University, Kotlářská 2, Brno 61137, Czech Republic. [3]Max-Plank-Institut für Radioastronomie, Max-Planck-Gesellschaft, Auf dem Hügel 69, Bonn 53121, Germany. [4]Astronomical Institute, Czech Academy of Sciences, Boční II 1401, Prague 141 00, Czech Republic. ✉e-mail: peissker@ph1.uni-koeln.de

may undergo a merging event due to the ongoing three-body interaction of the system with Sgr A*. The uncertain nature of the G objects can thus be resolved, at least in part, thanks to the binary system D9 whose imminent fate appears to be a stellar merger.

## Results

### Observations

Using archival data observed with the decommissioned near-infrared integral field unit (IFU) of Spectrograph for INtegral Field Observations in the Near Infrared (SINFONI, mounted at the Very Large Telescope)[18,19] in the H+K band (1.4−2.4 μm) between 2005 and 2019, we investigate the blue-shifted Brackett$\gamma$ (Br$\gamma$) emission of the source D9 (Fig. 1), which is part of the G-object population in the S cluster[16,17,20]. In addition, we include recent Enhanced Resolution Imaging Spectrograph (ERIS) observations carried out by the ERIS Team as part of the commissioning run in 2022[21]. For the analysis of the three-dimensional data cubes that consist of two spatial and one spectral dimension, we perform standard reduction steps (flat-fielding, dark, and distortion corrections). We obtain single barycentric and heliocentric corrected data cubes that are stacked for each year individually to construct a final mosaic of the entire S cluster region. Based on the best-fit Keplerian solution, we obtain an estimate of the periapse distance of the D9 system from Sgr A* of 29.9 mpc (0.75 arcseconds) adopting $M_{SgrA*} = 4 \times 10^6 M_\odot$ and 8 kpc for the mass and the distance of Sgr A*, respectively[22,23]. Furthermore, we find a close to edge-on orbital inclination of $(102.55 \pm 2.29)°$. With an eccentricity of 0.32 and a semi-

major axis of 44 mpc, D9 qualifies as an S cluster member with orbital parameters comparable to other S stars[3,24]. Due to the orbit of the B2V star S2 (S0-2) that intercepts with the trajectory of D9, we focus on the data set of 2019 to identify a continuum counterpart in the H and K band to the Br$\gamma$ line-emitting source.

### Magnitudes

To increase the photometric baseline, we incorporate Near-infrared Camera 2 (NIRC2, mounted at the Keck telescope) L band imaging data from 2019 to cover the near- and mid-infrared[25]. The science-ready data was downloaded from the Keck Observatory Archive[26]. Due to the high stellar density of the S cluster[27], dominant point spread function (PSF) wings are a common obstacle that hinders confusion-free detection of fainter objects such as G1[28], DSO/G2[29], or D9[20]. Therefore, we used an image sharpener on the continuum data of 2019 to reduce the impact of the challenging crowding situation in the S cluster (Supplementary Fig. 1 and Supplementary Table 1). With this procedure, we enhance fainter structures but preserve the photo- and astrometric aspects of the input data. To emphasize the robustness of the image sharpener, we invoke the contour lines of the input data as a comparison, as demonstrated in Fig. 1. Analyzing the displayed extinction corrected data (Supplementary Table 2), we find H −K = 1.75± 0.20 and K−L = 2.25 ± 0.20 colors for D9 suggesting photometric similarities with D2 and D23[20]. The latter two sources are believed to be associated with young T Tauri or low-mass stars[16,30,31]. Due to these photometric consistencies (Supplementary Fig. 2), we tested the hypothesis using a Spectral Energy Distribution (SED) fitter.

### Spectral energy distribution

The SED fitter[32] applies a convolving filter to the individual values to reflect on the response function of the instrument filter. Because the photometric system of SINFONI is based on the Two Micron All Sky Survey (2MASS) data base, we select the corresponding filters "2H" and "2K". For the NIRC2 MIR data, we use the United Kingdom Infrared Telescope (UKIRT) L' band filter because it is based on the Mauna Kea photometric system[33]. With these settings, the fitter compares models with the input flux (Fig. 2) where we limit the possible output that satisfies $\Delta\chi^2 \le 3$. These models represent young stellar objects (YSOs)

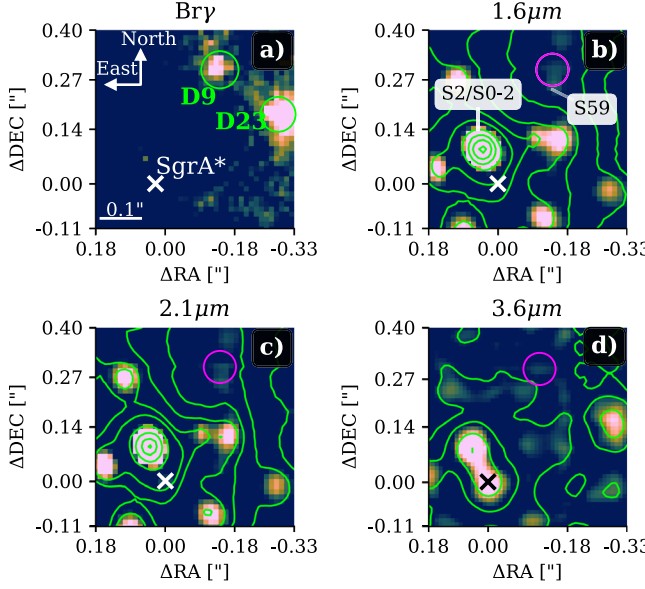

**Fig. 1 | Detection of the D9 system close to Sgr A* in 2019. a** Shows the Doppler-shifted Br$\gamma$ line map extracted from the H+K SINFONI data cube with a corresponding wavelength of 2.1646 μm (vacuum wavelength 2.1661 μm). **b**, **c** Shows the near-infrared H (1.6 μm) and K (2.1 μm) band data observed with SINFONI. **d** Denotes the mid-infrared L (3.76 μm) band observation carried out with NIRC2. Sgr A* is marked with a × , D9 is encircled in every plot. Due to its main sequence character, the marked close-by star S59 can only be observed in the H and K bands. On the contrary, the brightest K band source of the S cluster, S2/S0-2 can be observed in every shown infrared band. To increase contrast, an image sharpener is applied suppressing expansive point spread function (PSF) wings. To emphasize the astrometric robustness of the image sharpener, we adapt the lime-colored contour lines from the non-sharpened data. The contour line levels in **b** are at 10−80% of the peak intensity of S2, increasing in 5% steps. **c** The contour lines are set at 20−100% of the peak intensity of S2, separated by 10%. **d** The contour lines are set to 85%, 90%, 95%, and 100% of the peak intensity of S2. The labels of the axis indicate the distance to Sgr A* located at ΔRA = 0.00″ and ΔDEC = 0.00". In any plot shown, north is up, and east is to the left.

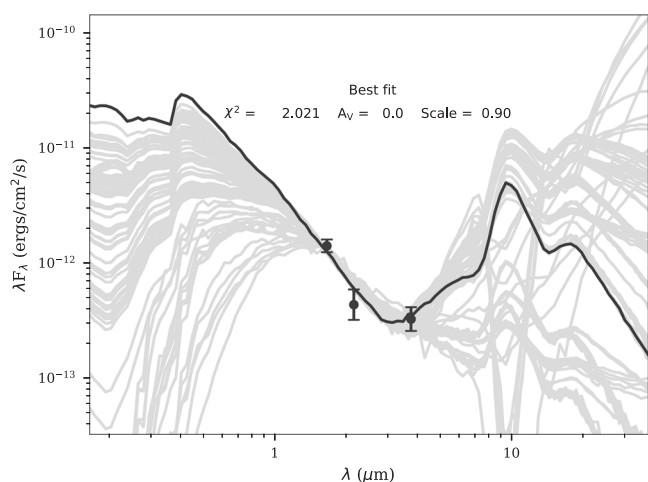

**Fig. 2 | Spectral energy distribution of the D9 system.** The extinction corrected data points refer to the flux density values in the H, K, and L band observed with SINFONI and NIRC2. We use $10^4$ individual models to find the best-fit of the data shown with gray lines. The final best-fit result is depicted with a black line. Based on the shown fit, the properties of the primary of the D9 binary system are derived and listed in Table 1. The uncertainties of the data points are estimated from the photometric variations along the source. Based on the reduced $\chi^2$ value of ~2, the displayed best-fit solution was selected.

and are composed of a stellar core, an accretion disk, and a dusty envelope. These typical components constitute a YSO and can be traced in the near- and mid-infrared parts of the spectrum. As input parameters, we used the H ($0.8 \pm 0.1$ mJy), K ($0.3 \pm 0.1$ mJy), and L ($0.4 \pm 0.1$ mJy) band flux densities estimated from the continuum detection presented in Fig. 1. Considering common YSO models, the H and K band emission traces the core components of the system, whereas the L band emission can be associated with a dusty envelope. Based on a photometric comparison with $10^4$ individual models, the best-fit of the SED fitter results in a stellar temperature of $1.2 \times 10^4$ K and a corresponding luminosity of approximately $93 L_\odot$, which are associated with a stellar mass of $2.8 \pm 0.5$ $M_\odot$ (see Table 1).

## Periodic pattern

While finalizing the analysis of D9, a pattern of radial velocity came to our attention. By inspecting the SINFONI mosaics that depict every observed night between 2005 and 2019, we found a clear periodic signal shown in Fig. 3 between $-67$ km/s and $-225$ km/s using the Doppler-shifted Br$\gamma$ emission line with respect to its rest wavelength at 2.1661 $\mu$m. A comparison of the periodic pattern of D9 with the Doppler-shifted Br$\gamma$ emission line of D23 demonstrates that the signal is not an artifact (Supplementary Fig. 3). From the orbital fit and the related inclination of $i = (102.55 \pm 2.29)°$, we know that D9 is moving on an almost edge-on orbit with a proper motion of $v_{prop} = 249.43 \pm 5.01$ km/s. Since S2 (S0-2) moves with a proper motion of almost 800 km/s[34], the comparable slow velocity of D9 implies that the intrinsic RV baseline $v_{base}$ of the system, estimated with $(v_{min} + v_{max})/2$, will not change significantly between 2005 and 2019. We normalize all observed velocities $v_{obs}$ to this baseline with $v_{obs}$-$v_{base}$

## Table 1 | Best-fit parameters of the D9 system

| Secondary Keplerian Parameter | |
|---|---|
| $P_{D9b}$ [year] | $1.02 \pm 0.01$ |
| $e_{D9b}$ | $0.45 \pm 0.01$ |
| $\omega_{D9b}$ [deg] | $311.75 \pm 1.65$ |
| $a_{D9b}$ [au] | $1.59 \pm 0.01$ |
| $i_{D9b}$ [deg] | $90.00$ |
| $m \sin(i_{D9b})$ [$M_\odot$] | $0.73$ |
| $RV_{off}$ [km s$^{-1}$] | $-29.19 \pm 3.00$ |
| $\chi^2_\nu$ | $0.31$ |
| $rms$ [km s$^{-1}$] | $16.38$ |
| Keplerian parameter for D9 orbiting Sgr A* | |
| $e_{D9a}$ | $0.32 \pm 0.01$ |
| $i_{D9a}$ [deg] | $102.55 \pm 2.29$ |
| $a_{D9a}$ [mpc] | $44.00 \pm 2.42$ |
| $\omega_{D9a}$ [deg] | $127.19 \pm 7.50$ |
| $\Omega_{D9a}$ [deg] | $257.25 \pm 1.61$ |
| $P_{D9a}$ [yr] | $432.62 \pm 0.01$ |
| Radiative transfer model | |
| $i_{intrinsic}$ [deg] | $75.0 \pm 19.0$ |
| R [$R_\odot$] | $2.00 \pm l0.13$ |
| $\log(L/L_\odot)$ | $1.86 \pm 0.14$ |
| $\log(T_{D9a}[K])$ | $4.07 \pm 0.05$ |
| $M_{D9a}$ [$M_\odot$] | $2.80 \pm 0.50$ |
| $M_{Disk}$ [$10^{-6}$ $M_\odot$] | $1.61 \pm 0.02$ |

We list the orbital parameters for the binary of D9 together with the motion of the system around Sgr A*. In addition, the best-fit stellar properties based on the SED fitter are included. The uncertainties of the binary parameter and the radiation transfer model are based on the reduced $\chi^2$. For the Keplerian elements, we use MCMC simulations to estimate the uncertainty range. Since the inclination of the secondary is assumed to be $i_{D9b} = 90°$, no uncertainty for $m \sin(i_{D9b}) = 0.73$ $M_\odot$ is given.

to obtain $v_{norm}$, which is the input quantity for the fit of the binary system performed with Exo-Stricker[35]. Due to the poor phase coverage before 2013, we split the data to perform an independent sanity check. The fit displayed in Fig. 3 resembles the epochs between 2013 and 2019, where we used a false-alarm probability of $10^{-3}$ similar to that used by ref. 14. The data baseline between 2005 and 2012 represents a non-correlated parameter to the Keplerian model of the binary provided by Exo-Striker, which is in agreement with the fit that is based on the epochs between 2013 and 2019 (Fig. 3). With a similar motivation, we incorporate the ERIS observations from 2022 that show a satisfactory agreement with the RV model and the expected LOS velocity of the binary, consisting of a primary and a secondary. Regarding the possible impact of a variable baseline $v_{base}$ (i.e., the LOS velocity $v_{obs}$ of D9 increases), we measure a difference of $\pm 15$ km/s between 2013 and 2019, which is consistent with the estimated uncertainty of $\pm 17$ km/s from the fit. We conclude that a variation of $v_{base}$ over the complete data baseline is inside the uncertainties and does not impact the analysis significantly. However, a forthcoming analysis of the binary system D9 should take this adaptation into account because it is expected that an alteration of the intrinsic LOS velocity will exceed the uncertainty range of the individual measurements of the periodic signal within the next decade.

In the subsequent analysis, we will refer to the primary as D9a, whereas the secondary companion will be denoted as D9b. With the binary orbiting Sgr A*, this three-body system is divided into an inner and outer binary. The inner binary describes D9a and D9b, while the outer one represents the D9 system orbiting Sgr A*.

The best-fit result includes an offset of $v_{base}$ with $RV_{off} = -29.19 \pm 3.00$ km s$^{-1}$ due to the eccentricity of the secondary $e_{D9b}$ of $0.45 \pm 0.01$, which causes an asymmetric distribution of the LOS velocity around the baseline. With this offset, we obtain $v_{mod} = v_{norm} + RV_{off}$ as displayed in Fig. 3. The related Keplerian parameters of the secondary orbiting its primary are listed in Table 1. From the fit but also evident in the periodic RV data points (Fig. 3), we find an orbital period for the secondary of $P_{D9b} = 372.30 \pm 3.65$ days $= 1.02 \pm 0.01$ yr, which can be transferred to a total mass of the system of about $M_{bin} = 3.86 \pm 0.07$ $M_\odot$, considerably above the derived D9 (i.e., the primary) mass of $M_{D9a} = 2.8 \pm 0.5$ $M_\odot$. The difference in mass for $M_{D9a}$ and $M_{bin}$ cannot be explained solely by the uncertainty range. However, inspecting $m \sin(i_{D9b}) = 0.73$ $M_\odot$ and the assumed inclination of the secondary of 90° results in the maximum mass of the companion. The assumed inclination of the secondary is motivated by an almost edge-on orbit of D9 (Table 1). Although the circumprimary disk does not necessarily have to be aligned with the orbit of the binary as is found for T-Tauri systems[36], surveys of Herbig Ae/Be stars suggest a tendency towards a coplanar arrangement[37]. Assuming that the orbit of the secondary is approximately aligned with the circumprimary disk with an intrinsic inclination of the primary D9a of $i_{intrinsic} = (75 \pm 19)°$ (Table 1), we are allowed to transfer the related uncertainties to $m \sin(i_{D9b})$. Following this assumption, we find a mass for the secondary of $M_{D9b} = 0.73 \pm 0.14$ $M_\odot$ consistent with the derived primary mass of $M_{D9a} = 2.8 \pm 0.5$ $M_\odot$ and the total mass $M_{bin} = 3.86 \pm 0.07$ $M_\odot$ of the system.

## Discussion

### Radiation mechanism

Taking into account the periodic variation of Br$\gamma$ emission, we want to highlight three different scenarios as a possible origin of the periodic Br$\gamma$ signal.

Firstly, the emission of the Br$\gamma$ line is solely the result of a combination between the gaseous accretion disk and stellar winds of the primary[38,39]. In this scenario, the secondary disturbs this emission by its intrinsic Keplerian orbit around the primary.

Secondly, a possible origin of the Br$\gamma$ line could be the presence of a circumbinary disk around the D9 binary system enveloping the primary and the secondary. In this case, the interaction between the

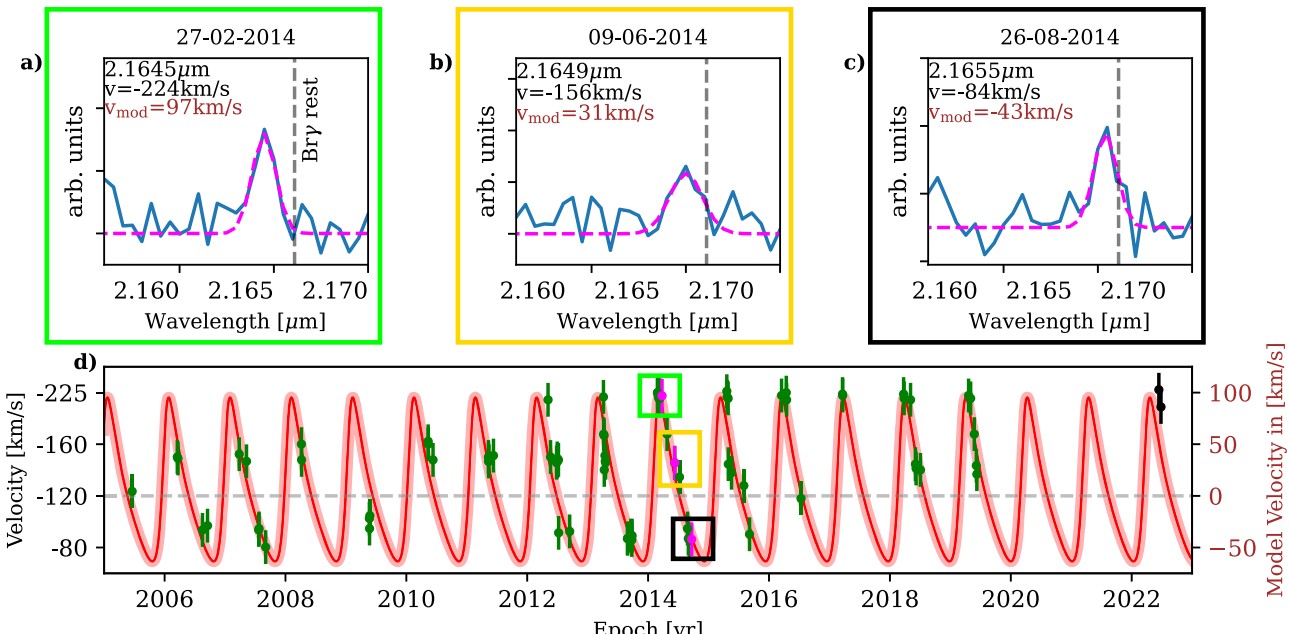

**Fig. 3 | Radial velocity of D9 between 2005 and 2022 observed with SINFONI and ERIS. a**–**c** We display three selected nights to show the variable Brγ emission line with respect to the rest wavelength at 2.1661 μm. The top three plots correspond to the same colored boxes as in the radial velocity evolution model shown in (**d**). We have indicated the exact data point using magenta color. **d** The SINFONI data is indicated in green, and the two ERIS observations from 2022 are highlighted in black. Due to the decommissioning, no high-resolution spectroscopic data are available between 2020 and 2021. In addition, the usual observation time for the Galactic center at Cerro Paranal (Chile) is between March and September, which explains the limited phase coverage. All data points in the radial velocity (**d**) correspond to a single night of observation. The velocities in the left *y* axis are related to the observed blue-shifted Brγ emission lines. Due to data processing, these values are shifted and arranged to an estimated zero-velocity baseline (see the right *y* axis). The uncertainties of the individual data points are calculated from the root-mean-square (RMS) deviation (see Table 1).

primary with the secondary allows inward gas streams from the circumbinary disk resulting in the observed periodic Brγ line[40].

The third and foremost plausible scenario is the interaction between two accreting stellar objects. It is well-known that especially Herbig Ae and T-Tauri stars exhibit prominent Brγ emission lines associated with accretion mechanisms[41,42]. For instance, a radial shift of the accretion tracer has been observed for the DQ Tau binary system[43]. It has been proposed that this resonance-intercombination may be explained by stellar winds of the secondary[44]. Due to Keplerian shear, line photons can escape the optically thick material and produce the RV pattern, as observed for the D9 binary system[45].

### Stellar types of primary and secondary

Considering the presence of a primary and its companion, it is suggested that stellar winds interact with the Brγ emission of the accretion disk(s) of the binary system[38,46] that gets periodically disturbed by the presence of the secondary[47,48]. Alternatively, the Brγ emission line is produced by both the primary and secondary as it is observed for the Herbig Ae star HD 104237 with its T-Tauri companion[49]. Comparing $M_{D9a}$ with the total mass of $M_{bin}$ = 3.86 $M_{\odot}$ of the system suggests that the secondary does not contribute significantly to the photometric measurements analyzed in this work. If it were not the case, the estimated mass for the primary of the D9 system of $M_{D9a}$ = 2.8 ± 0.5 $M_{\odot}$ would be lower, while $M_{D9b}$ = 0.73 ± 0.14 $M_{\odot}$ should be increased. Considering the estimated mass of the primary $M_{D9a}$ and the fixed upper limit of $M_{bin}$ based on the observed period, the secondary can be classified as a faint low-mass companion, suggesting a classification as a T-Tauri star[50]. Considering the stellar mass, radius, and luminosity of the primary (Table 1), the system may be comparable to the young Herbig Ae/Be star BF Orionis, which is speculated to also have a low-mass companion[51]. On the basis of observational surveys, it is intriguing to note that most Herbig Ae/Be stars exhibit an increased multiplicity rate of up to 80%[37,52]. Another result of the radiative transfer

model is the relatively small disk mass $M_{Disk}$ of (1.61 ± 0.02) × $10^{-6}$ $M_{\odot}$, which could be interpreted as an indicator of the interaction between D9a and its low-mass companion D9b. Possibly, this ongoing interaction, but most likely the stellar winds of the S stars[53], will disperse the disk of D9a in the future[54–57]. Using the derived luminosity and stellar temperature of D9a together with the evolutionary tracks implemented in PARSEC[58], we estimated the age of the system of $2.7^{+1.9}_{-0.3} \times 10^6$ yr.

### Migration scenario

A potential migration scenario has been proposed by ref. 59 and can be described as the triple-system hypothesis. In this scenario, a triplet system migrates towards Sgr A*[60–62], where the two companions are captured to form a binary. It is possible that the third companion may be ejected from the cluster and subsequently become a hyper-velocity (HV) star, as postulated by refs. 63,64. A consequence of the disruption of the initial triplet is the resulting high eccentricity of the captured binary system close to unity[65]. Since the derived outer eccentricity of the D9 system is $e_{D9a}$ = 0.32 ± 0.01 (Table 1), we consider a migration channel different from the triple-system hypothesis. As proposed by refs. 60 and [61], molecular clouds can migrate towards the inner parsec and consequently close to Sgr A*. Speculatively, the D9 system could have formed during such an inspiral event. An additional implication based on the age estimate is the presumably evaporated circumbinary disk that enveloped the primary and secondary. The authors of ref. 66 found that the timescales for dismantling the circumbinary disk scale with the separation between the primary and secondary. The relation can be formulated with $t_{dis.time} \leq 10^6$ yr$< t_{D9a, age} = 2.7^{+1.9}_{-0.3} \times 10^6$ yr. The former relation is strengthened by the analysis of ref. 67 who found that photoevaporative winds decrease the lifetime of the circumbinary disk as a function of distance. Independent of the stellar wind model, the author of ref. 67 found that circumbinary disks evaporate between ~1–10 × $10^6$ yr providing an explanation for the low disk mass of

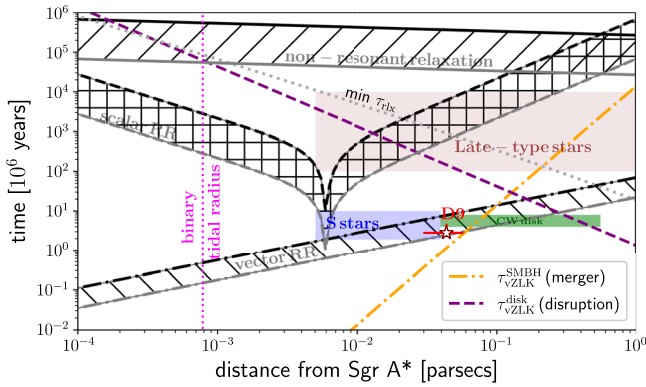

**Fig. 4 | Distance and age of D9 in the context of basic dynamical processes and stellar populations in the Galactic center.** In terms of the semi-major axis, D9 is positioned in the outer part of the S cluster, close to the innermost part of the clockwise (CW) disk of OB/Wolf-Rayet stars. With its estimated age of $2.7^{+1.9}_{-0.3} \times 10^6$ yr, its orbit around Sgr A* can just be under the influence of the fast vector resonant relaxation (RR; shaded area stands for the vector resonant relaxation of a $1\,M_\odot$ star and a $10\,M_\odot$ star represented by the top and the bottom lines, respectively). However, the scalar resonant relaxation (RR) and the non-coherent two-body relaxation have not had sufficient time to affect significantly the angular momentum and the orbital energy of the D9 system yet. Hence, D9 as a binary system is currently stable against the tidal disruption by Sgr A* (vertical dotted magenta line denotes the binary tidal radius). A similar conclusion can be drawn with regard to the minimum relaxation time min $\tau_{\rm rlx}$ resulting from the dark cusp (illustrated by the orange dotted line). In addition, the von Zeipel-Lidov-Kozai (vZLK) mechanism that involves the SMBH-D9-CW disk ($\tau_{vZLK}^{\rm disk}$; dashed purple line) operates on a long timescale to cause the tidal disruption of the binary. On the other hand, in the hierarchical setup where the inner D9 binary orbits the SMBH, the corresponding vZLK timescale is comparable to the age of D9, which implies a likely merger (orange dash-dotted line).

$(1.61 \pm 0.02) \times 10^{-6}\,M_\odot$ found for the D9 binary system. Between 2005 and 2022, the D9 binary system has remained stable in the gravitational potential dominated by Sgr A*. This is evident from the observable periodic RV signal for almost 20 years. The conditions for the dynamical stability of the binary can be extracted directly from the Keplerian orbital fit and binary mass estimate by calculating the tidal (Hill) radius.

For the periapse distance $r_{\rm p}$ of approximately 30 mpc corresponding to 6200 AU, we find the tidal (Hill) radius for D9 of $r_{\rm Hill} = r_{\rm p}(M_{\rm bin}/3M_{\rm SgrA*})^{1/3} = 42.4$ AU. The effective orbital radius of the inner binary system is $r_{\rm eff} = 1.26 \pm 0.01$ AU using the Keplerian orbital parameters for the secondary listed in Table 1. Therefore, the system remains in a stable, mildly eccentric orbit around Sgr A*, and it can be further described as a hard binary. This is expected since the evolution of the outer orbit of the system D9-Sgr A* is dominated by the gravitational potential of the SMBH. However, because of its age and potential interaction with the dense environment, the question of binary destruction timescales should be addressed. It is plausible that the inner system D9a-D9b will actually become even harder and the components will eventually merge[68]. This is due to the interaction of the D9 system with Sgr A*, which acts as a distant massive perturber that alters the orbital parameters through the von Zeipel-Lidov-Kozai (vZLK) mechanism[69-71]. Due to the young age of the binary system and, therefore, the short time in the S cluster (compared to the evolved stars), we will focus in the following section on the vZLK and other effects induced by the dark cusp of the S cluster.

## Dynamical processes and stellar populations

The lifetime of the D9 system with its estimated age of $2.7^{+1.9}_{-0.3} \times 10^6$ yr and the semi-major axis of about 44 mpc can be compared with basic dynamical processes and their timescales as well as with other known

stellar populations in the central parsec in the distance-timescale plot. Such a plot (see e.g., ref. 72) can be used to infer which dynamical processes can be relevant for the current and the future orbital evolution of D9 at a given distance. We use the timescales for the two-body non-resonant relaxation $\tau_{\rm NR}$, scalar and vector resonant relaxation $\tau_{\rm RR}^{\rm s}$ and $\tau_{\rm RR}^{\rm v}$, respectively, and the vZLK mechanism driving inclination-eccentricity oscillations taking place on the vZLK timescale $\tau_{\rm vZLK}$. In Fig. 4, we show the D9 system (red star), the timescales related to the dynamical processes, and the relevant stellar populations identified in the inner parsec: S cluster, clockwise (CW) disk, and late-type stars. For most of the timescales (non-resonant, scalar, and vector resonant relaxations), we need an estimate for the number of stars inside the given distance $r$ from Sgr A*, $N(< r)$. For this purpose, we use the power-law mass density profile $\rho(r) = 1.35 \times 10^5 (r/2\,{\rm pc})^{-1.4}\,M_\odot\,{\rm pc}^{-3}$, whose power-law index is adopted from ref. 73 and the normalization coefficient is determined so that $M(<2\,{\rm pc}) = 2M_{\rm SgrA*}$, i.e. twice the Sgr A* mass at the influence radius. We see that for the inferred age of D9, none of the relaxation processes is fast enough to change significantly the angular momentum magnitude, i.e. the eccentricity. Hence, the D9 binary is stable against disruption by Sgr A* at the corresponding tidal radius $r_{\rm t}$ of about $161(a_{\rm D9b}/1.59\,{\rm AU})(M_{\rm SgrA*}/4 \times 10^6\,M_\odot)^{1/3}$ $(M_{\rm bin}/3.86\,M_\odot)^{-1/3}$ AU $\simeq 0.78$ mpc, for which the orbital eccentricity of $e \simeq 1 - r_{\rm t}/a = 0.98$ would be required. Apart from non-resonant and scalar resonant relaxation processes, such a high eccentricity of the D9 orbit around Sgr A* cannot be reached via the vZLK oscillations, where we consider Sgr A*−D9 as an inner binary and the CW disk as an outer perturber with the mass of $M_{\rm disk} \lesssim 10^4\,M_\odot$. With the mean distance of the disk $r_{\rm disk}$ of about 0.274 pc from D9, the corresponding vZLK cycle timescale is given by,

$$
\begin{aligned}
\tau_{\rm vZLK}^{\rm disk} &= 2\pi \left(\frac{M_{\rm SgrA*}}{M_{\rm disk}}\right)\left(\frac{r_{\rm disk}}{a_{\rm D9a}}\right)^3 P_{\rm D9a} \\
&= 2.6 \times 10^8 \left(\frac{M_{\rm SgrA*}}{4 \times 10^6\,M_\odot}\right)\left(\frac{M_{\rm disk}}{10^4\,M_\odot}\right)^{-1}\left(\frac{r_{\rm disk}}{0.274\,{\rm pc}}\right)^3 \times \\
&\quad \times \left(\frac{a_{\rm D9a}}{0.044\,{\rm pc}}\right)^{-3}\left(\frac{P_{\rm D9a}}{432.35\,{\rm years}}\right)\,{\rm yr},
\end{aligned}
\tag{1}
$$

which is two orders of magnitude longer than the lifetime of D9 (see also Fig. 4 for the radial dependency of $\tau_{\rm vZLK}^{\rm disk}$). In Eq. (1), we adopted the notation of the D9 orbital parameters as summarized in Table 1.

When we concentrate instead on the other hierarchical three-body system−the inner D9 binary and the outer binary D9-Sgr A*, the inner binary components undergo the vZLK inclination-eccentricity cycles. The corresponding vZLK timescale then is,

$$
\begin{aligned}
\tau_{\rm vZLK}^{\rm SMBH} &= 2\pi \left(\frac{M_{\rm bin}}{M_{\rm SgrA*}}\right)\left(\frac{a_{\rm D9a}}{a_{\rm D9b}}\right)^3 P_{\rm D9b} \\
&= 1.1 \times 10^6 \left(\frac{M_{\rm bin}}{3.86\,M_\odot}\right)\left(\frac{M_{\rm SgrA*}}{4 \times 10^6\,M_\odot}\right)^{-1}\left(\frac{a_{\rm D9a}}{0.044\,{\rm pc}}\right)^3 \times \\
&\quad \times \left(\frac{a_{\rm D9b}}{1.59\,{\rm AU}}\right)^{-3}\left(\frac{P_{\rm D9b}}{1.02\,{\rm years}}\right)\,{\rm yr},
\end{aligned}
\tag{2}
$$

which is within the uncertainties comparable to the age of D9. In Eq. (2), we adopted the notation of the parameters of both the D9 orbit around Sgr A* and the binary orbit as summarized in Table 1. Hence, the system appears be detected in the pre-merger stage. As the eccentricity of the D9 binary will increase during one vZLK timescale, the strong tidal interaction between the components during each periastron will perturb the stellar envelopes significantly, which will plausibly lead to the merger of both components once they are significantly tidally deformed[74]. Such a merger process is first accompanied by the Roche-lobe overflow of the stellar material from one of

the components and then a subsequent merger of the stellar cores (see e.g., ref. [75]). At the same time, the common envelope is progressively inflated to several thousand Solar radii. As it cools down, the infrared excess increases considerably. In this way, some or all of the G objects observed in the Galactic center could be produced, and the D9 system would represent a unique pre-merger stage, which is also hinted by the smaller near-infrared excess in comparison with other G objects[17].

## Fate of the binary

Considering the age and presence of the binary system in the S cluster, we will examine the impact of the dark cusp. Old and faint stars have migrated into the S cluster from a distance of a few parsecs[6] and might alter the orbits of the young and bright cluster members[7,68,76]. With the detection of the binary system D9, we convert its stellar parameters (Table 1) and age of $T_{D9a} = 2.7^{+1.9}_{-0.3} \times 10^6$ yr to a lower limit for the minimum two-body relaxation timescale of $\min t_{rlx} = 4.8(M_{SgrA*}/M_{bin})(a_{D9b}/a_{D9a})T_{D9a}$ resulting in about $874 \times T_{D9a}$ yr[76], equivalent to approximately $10^9$ yr exceeding the lifetime of the binary by three orders of magnitude. This suggests that the dark cusp does not have any significant imprint on the D9 system independent of its time in the cluster. Given that the assumed inclination is a geometrical parameter contingent upon the observer, it is reasonable to conclude that it will have, such as the dark cusp, no impact on the dynamical evolution of the binary system. We will now examine the evolutionary path that is described by the vZLK mechanism where D9 is the inner binary and D9-Sgr A* represents the outer binary[74]. For this hierarchical setup, the vZLK timescale is $\tau^{SMBH}_{vZLK} = 1.1 \times 10^6$ yr, see Fig. 4 and Eq. (2), which is comparable with the approximate lifetime of the binary of $T_{D9a} = 2.7 \times 10^6$ yr. It is reasonable to assume that the ongoing interaction between the primary, secondary, and Sgr A* is reflected in altering the eccentricity of the D9 binary, which very likely results in a merger. This supports the idea that the G-object population[17] has a contribution from recently merged binary systems, as proposed by ref. [16]. Considering the vZLK timescale $\tau^{SMBH}_{vZLK}$ of about $10^6$ yr and the age of D9 of $2.7^{+1.9}_{-0.3} \times 10^6$ yr, the system could have migrated to its current location and may soon merge to become a G-object. D9 thus offers a glimpse on one potential evolutionary path of the S stars. Taking into account that the bright and massive B-type S stars with an average age of $6 \times 10^6$ yr[8,9] may have formed as binary systems[12], it is suggested that these young S cluster members might have lost their putative companions in the immediate vicinity of Sgr A* assuming an ex-situ formation. In refs. 11 and 14, the authors explored the probability density for the young stars in and outside the S cluster. The authors propose that the probability of a binary system is significantly higher outside the central arcsecond (≥72% compared to ≤17% at 68% confidence interval). If we consider the recent detection of the new G-object X7.2[77], we estimate with $R = N_B/(2N_m)$ a binary fraction of the central 0.1 pc to be ~10 % using the Ansatz of ref. [16], where $N_B = 86$[20,77] represents the assumed number of binaries and $N_m = 478$[16] the amount of low-mass stars in the S cluster using the initial mass function derived by ref. [8]. This implies that the majority of expected binaries in the S cluster should be among the G objects[20].

Regardless of the formation or migration scenarios, we can estimate that the B-type stars of the S cluster reside in their environment for at least $1.1 \times 10^6$ years due to the absence of their expected companion stars[11,12]. The estimated vZLK timescale is compatible with the predicted decrease of binaries for a possible star-formation episode in the Galactic center $6 \times 10^6$ yr ago[8,74]. This suggests that the vZLK mechanism may be the driving force of the decrease in binary fraction in the dense S cluster[7,68,78].

## Alternative explanations

The number of detected binaries in the Galactic center is surprisingly low. Only five confirmed binaries have been found, which is, considering an approximate number of stars in the NSC of ~10[61], a negligible fraction of the overall population (Supplementary Table 3). Although the multiplicity fraction in the NSC should be higher[74,78], other possible scenarios that explain the periodic RV pattern displayed in Fig. 3 should be taken into account. One possible alternative explanation for the periodic variations of RV could be stellar pulsations[79]. This scenario was initially used to explain the photometric variability of IRS 16SW[80,81]. However, it was later confirmed that the Ofpe/WN9 star IRS 16SW is indeed a massive binary by conducting IFU observations with SINFONI[82] analyzing the Brγ emission line. Considering the binary period of the D9 system of ~372 days, stellar pulsations are rather unlikely, since they occur on daily timescales[83]. Alternatively, the Brγ emission could be related to the rotation of the accretion disk of D9. Although ionized hydrogen and disk winds are associated with YSOs[38,46], the dimensions of the disk itself and the spectral resolution of the instrument pose a strong constraint on the detectability of the system.

# Methods

## Age of the system

For an age estimate of the D9 binary system, we use the temperature and radius listed in Table 1 with stellar evolutionary tracks from PARSEC[58]. Considering the low mid-infrared flux in the L band of $0.4 \pm 0.1$ mJy compared to the K band of $0.8 \pm 0.1$ mJy, questions the proposed classification for D9 as a candidate Class I YSO as suggested by ref. [17]. Taking into account the derived stellar mass of the system in combination with the hydrogen emission line, alternative explanations are required to classify the binary system. As outlined before, it is known that the Brγ line is a tracer for accretion disks of Herbig Ae/Be stars[39]. Similar to Herbig Ae/Be surveys[84], we use the PARSEC isochrones[58] to estimate the age of D9 (Fig. 5). We find an age of the D9 system of $2.7^{+1.9}_{-0.3} \times 10^6$ yr (Fig. 5), which is, in combination with the high binary rate[37,52,84], typical for Herbig Ae/Be stars. This age estimate implies an ex-situ formation scenario because the dominant winds of the massive stars inside the S cluster would have photoevaporated the required star-formation material in the first place[53,85]. The stellar evolution model is in agreement with common stellar parameters of Herbig Ae/Be stars[84,86] that are derived from the Gaia Data Release 2[87,88].

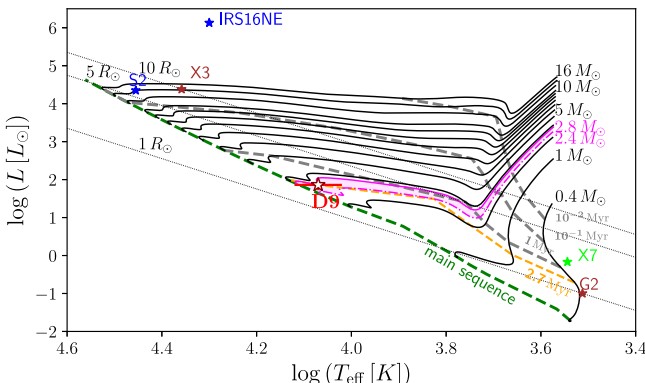

**Fig. 5 | Hertzsprung-Russel diagram using the evolutionary tracks based on the PARSEC stellar evolution model.** The D9 binary system is indicated by a red star with the corresponding error bars in the temperature-luminosity plot. The magenta-shaded area depicts the range of the masses of stars ($2.4$–$2.8\,M_\odot$), whose stellar evolution is consistent with the location of the D9 source at the time of $2.4$–$4.6 \times 10^6$ yr. The orange-dashed line represents the isochrone corresponding to 2.7 million years. For comparison, we implement known sources of the Galactic center, such as the putative high-mass YSO X3[98], the bow-shock source X7[77,99], dusty S cluster object G2[100], and the massive early-type stars S2[9] and IRS16NE[101].

## Keplerian orbit

Using the well-known orbit of S2 (S0-2)[89,90], we determine the position of Sgr A*. Since the intrinsic proper motion of Sgr A*, $v_{prop, SgrA^*}$, is only a fraction of a pixel per epoch[91] and thus several orders of magnitude smaller than the distance to D9, we neglect this velocity term. The rejection of $v_{prop, SgrA^*}$ is motivated by the typical astrometric uncertainties of ±12.5 mas that exceed the intrinsic proper motion of Sgr A* with $v_{prop, SgrA^*} = 0.3$ mas/yr. From the fixed position of Sgr A*, we use the astrometric information of D9 to derive a related Keplerian orbital solution. We incorporate the LOS velocity of D9 using the estimated baseline of ~150 km/s and a corresponding uncertainty range of ±15 km/s. Comparing the statistical significance of the Keplerian fit with and without the LOS velocity results in a difference of almost one magnitude for the reduced $\chi^2$. We estimate $\chi^2_{red}$ to be about 10 for the sole astrometric measurements while we find a robust fit for $\chi^2_{red}$ of approximately 2 by maximizing the parameter space, that is, including the LOS velocity. With a mass of $M_{SgrA^*} = 4 \times 10^6\,M_\odot$ for Sgr A*[22,23], we display the resulting Keplerian orbit in Fig. 6 and list the corresponding orbital elements in Table 1. As is evident from the plot displayed in Fig. 6, D9 moves on the descending part of its Keplerian orbit, which results in the mentioned slow velocity. Intriguingly, the relative location and its intrinsic velocity of D9 with respect to Sgr A* ensure a confusion-free detection of the binary system. Detecting the binary would most likely be hindered if it was in its ascending part of the orbit.

## Statistical analysis

The Limited-memory Broyden, Fletcher, Goldfarb, and Shannon box constraints (L-BFGS-B) algorithm forms the basis of the Keplerian fit[92,93]. The Keplerian fit relies on the L-BFGS-B algorithm, which is an iterative method that identifies free parameters within a given range and aims to minimize the gap between the data points and the priors (i.e., initial guess). The Keplerian equations of motion describe the model underlying the algorithm. The algorithm iteratively finds the orbital solution that best fits the data points with high accuracy, i.e., the minimized $\chi^2$.

The best-fit parameters are then used as a prior for the Markow-Chain-Monte-Carlo (MCMC) simulations. The MCMC algorithm was used by the implementation of the emcee PYTHON package developed by ref. 94. When inspecting the distribution of the measured data points, it is evident that the D9 system moves with a comparable slow velocity in the S cluster, which translates into an almost (projected) linear motion. Hence, it is not entirely unexpected that the MCMC simulations are in high agreement with the best-fit results of the Keplerian approximation (Table 2). We can conclude that the orbital solution presented in Table 2 is robust and should provide a suitable basis for future high-angular resolution observations.

## Uniqueness of the IFU data points

The line maps of the three-dimensional data cubes observed with SINFONI and ERIS act as a response actor, which is interpreted as a measure of the influence of nearby sources and the imprint of the background. It is important to note that sporadic background fluctuations do not result in a line map emission counterpart. In other words, the line emission with spatially limited origin (i.e. noise) does not produce a (compact) line map signal comparable to, e.g., G2[29]. This is due to the flux required to produce a signal above the sensitivity level of the detector. Vice versa, only spatially extended emission with sufficient line emission produces a spectroscopic signal (Supplementary Fig. 4 and Supplementary Fig. 5). This interplay between line emission and line maps reduces the chance of detecting false positives of any kind. Mathematically speaking, the mentioned interplay between the two parameter spaces (spatial and spectroscopic) of detecting a real signal is a necessary condition. In this sense, one cannot claim the existence of a source based on one parameter space.

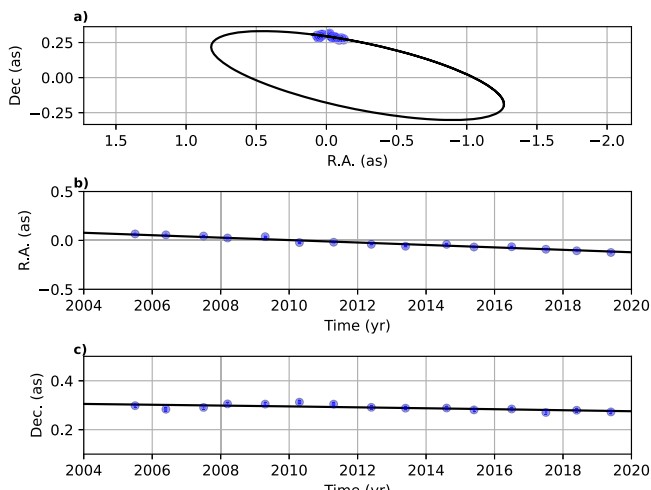

**Fig. 6 | Keplerian orbit of the D9 system. a** The projected on-sky trajectory of the D9 binary system is shown. **b, c** Shows the R.A. and DEC. position as a function of time. **b, c** The low proper motion is eminent. Every blue-colored data point in this figure is related to one observational epoch. From this plot and the related inclination of $i_{orb} = (102.55 \pm 2.29)°$, it is suggested that the trajectory of the binary system is close to edge-on. The size of the blue data points is related to the astrometric uncertainty of ±0.006 arcsecond (as).

**Table 2 | Comparison of best-fit Keplerian approximation and MCMC simulations**

| Parameter | Best-fit | MCMC | Standard deviation |
|---|---|---|---|
| $a_{D9a}$ [mpc] | 44.00 | 45.55 | 1.15 |
| $e_{D9a}$ | 0.32 | 0.31 | 0.01 |
| $i_{D9a}$ [°] | 102.55 | 103.30 | 1.14 |
| $\omega_{D9a}$ [°] | 127.19 | 130.96 | 8.02 |
| $\Omega_{D9a}$ [°] | 257.25 | 258.40 | 1.71 |
| $t_{closest}$ [years] | 2309.13 | 2315.83 | 7.01 |

Since the standard deviation does not satisfactorily reflect the astrometric precision that can be achieved with SINFONI, we will use the standard deviation of the combined MCMC posteriors. These orbital elements are related to the outer binary system D9-Sgr A*. We refer to ref. 29 for a detailed explanation of the background fluctuations of the SINFONI data.

Taking into account the Keplerian orbit of D9 further reduces the probability of a false positive, which occurs only at the expected orbit position, by several magnitudes. Refs. 95 and [27] calculated the probability of detecting an artificial source on a Keplerian orbit to be in the range of a fraction of a percent. This can only be considered an upper limit because the probability relates to a time span of 5 years and covers solely astrometric data. In Fig. 7, we show an overview of selected epochs to demonstrate the interplay between the observed Brγ emission line and the line maps. These line maps are created by selecting a wavelength range of ~0.0015 μm, which corresponds to three channels in total (out of 2172 channels in total). A crucial pillar of the binary detection presented in this work is the analysis of individual nights observed with SINFONI and ERIS. Therefore, it is expected that the quality of the data will differ not only due to variable weather conditions but also to the number (i.e., on-source integration time) of observations executed at the telescope (Fig. 7). Of course, the impact of these boundary conditions is reduced by stacking individual cubes, as has been done for the analysis presented, for example, in refs. 20,29,96. Since the RV signal of the D9 system changes on a daily basis, stacking these single night data cubes affects the signal-to-noise ratio (SNR) of the Brγ line emission of the D9 system (Supplementary Fig. 4). For example, the signal-to-noise ratio for the stacked 2019

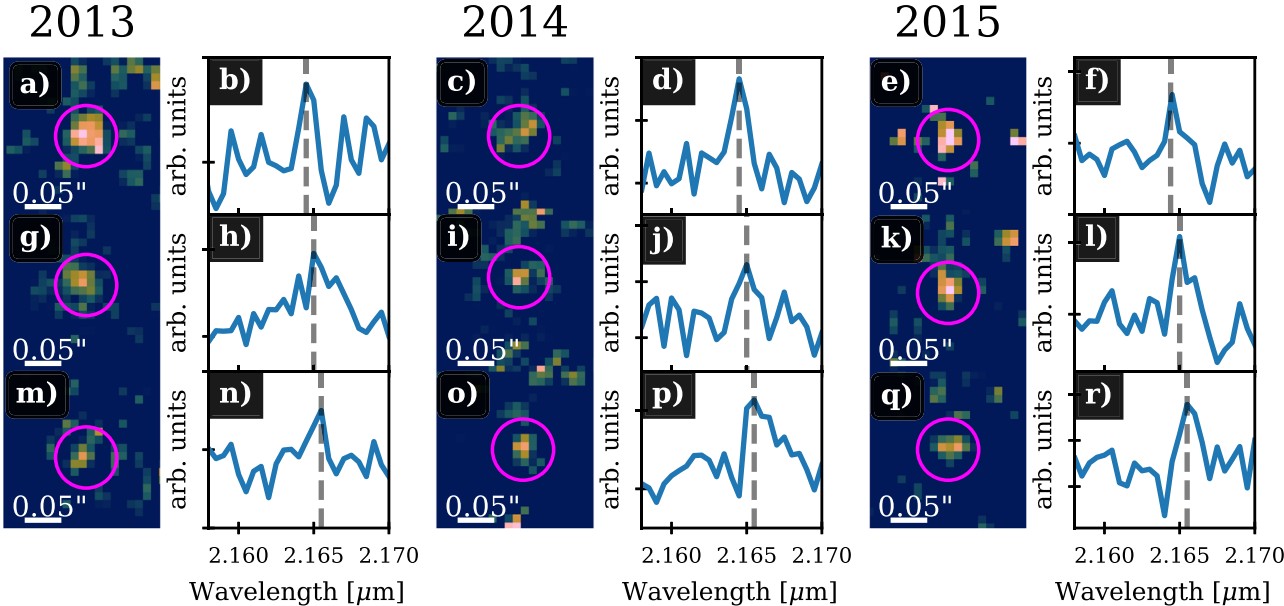

**Fig. 7 | Doppler-shifted Brγ line of D9 and the related line maps representing the magenta-marked emission. a, c, e, g, i, k, m, o,** and **q** show SINFONI line maps of the binary system D9. In these subplots, D9 is marked with a magenta-colored circle. **b, d, f, h, j, l, n, p,** and **q**, we apply a local background subtraction of the surrounding gas to the presented spectra. The successful subtraction of the background is evident in the absence of the prominent Brγ peak at 2.1661 μm[102,103]. The shown spectra shows the evolution of the line over one year. The normalized Brγ velocity $v_{norm}$ in 2013 is -66 km/s (**b**), 3 km/s (**h**), and −72 km/s (**n**). In 2014, $v_{norm}$ is -68 km/s (**d**), 3 km/s (**j**), and −71 km/s (**p**). In 2015, we estimate $v_{norm}$ to be -72 km/s (**f**), 1 km/s (**l**), and −67 km/s (**r**).

SINFONI data cube with an on-source integration time of almost 10 hours is 20, while two cubes from a single night in 2019.43 show an average SNR of ~5. Although detection of the D9 binary system would benefit from using the data cubes that include all annual observations, an analysis of the periodic RV signal would be hindered.

### ERIS data

The ERIS data analyzed in this work are part of the science verification observations carried out in 2022 by the PI team. To reduce the data, we use the ESO pipeline[97] that applies the standard procedure (dark, flat, and distortion correction). Furthermore, the data are part of a preliminary analysis of the Galactic center with ERIS[21]. The authors of ref. 21 report a superior performance compared to SINFONI, which can be confirmed as shown in Fig. 8. Although the on-source integration time is only 1200 seconds for each night, we find an SNR of almost 6 for the Doppler-shifted Brγ emission line of the D9 binary system. In both data sets shown, we detect D9 close to D23 without confusion comparable to the SINFONI observations displayed in Fig. 1 at the expected wavelength (Fig. 3). Due to the distance between D9 and D23 in 2022, both sources will be affected by interference in forthcoming observations of the S cluster.

### Radial velocity fit

For the spectrum that is used to extract the related LOS velocity, we subtract the underlying continuum by fitting a polynomial to the spectroscopic data. Line maps are constructed in the same way directly from the three-dimensional data cubes (Fig. 1). Using an aperture with a radius of 25 mas, the extracted spectrum of D9 reveals a velocity range between −67 km/s and −225 km/s (Supplementary Tables 4–6) on the investigated data baseline with a corresponding average LOS velocity of $v_{LOS} = −153.72$ km/s and a measured uncertainty of 16.38 km/s (Table 1). If the source is isolated, we use an annulus for a local background subtraction[31]. In any other case, we select an empty region 0.1" west of S59 (Fig. 1). Subtracting the baseline $(v_{min} + v_{max})/2$ from the individual velocity values normalizes the distribution. With this arrangement of the observed RV, we used the tool Exo-Striker[35] to

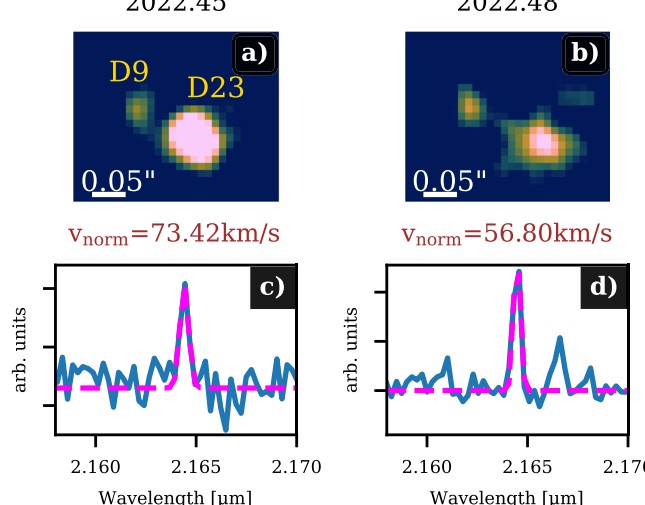

**Fig. 8 | Observations of the D9 binary system in 2022 with ERIS. a, b** The Brγ line maps observed with ERIS in 2022 are shown. Both subplots display the binary system D9 and the close-by source D23. For visualization purposes, we apply a 40 mas Gaussian kernel to these line maps. **c, d** Show the related spectrum where we indicate the normalized RV velocity $v_{norm}$. Including the offset measured by Exo-Striker of -29 km/s, these velocities are displayed as black data points in Fig. 3.

fit the related velocities, which resulted in the binary orbital parameter listed in Table 1 and the Keplerian fit of the secondary trajectory displayed in Fig. 3. The model predicts a secondary on an elliptical orbit around the primary, which further results in an RV offset of -29 km/s. This offset is added to the normalized velocities. As shown in Fig. 3, the final normalized LOS velocity is around −120 km/s. The reduced chi-square is $\chi^2 = 0.31$, which implies a significant agreement between the data and the fit. Due to the extended data baseline of 15 years (Supplementary Tables 7–9), we established an independent sanity check

to reflect the satisfactory agreement of the observed RV and the fit. For this, we split the data and limit the fit to the epochs between 2013 and 2019. Hence, the epochs before 2013 represent a non-correlated parameter to the Keplerian model provided by Exo-Striker with an average LOS velocity of $v_{LOS*} = -147$ km/s. The difference between the average $v_{LOS}$ and $v_{LOS*}$ is expected due to the phase coverage and the intrinsic LOS velocity of D9. We note that both averaged velocities are within the estimated uncertainties. It is also notable that the independent RV data before 2013 and after 2019 match the derived periodic model of the D9 binary system.

## Data availability
The open-access SINFONI and ERIS raw data can be downloaded from https://archive.eso.org/eso/eso_archive_main.html using the related observation ID indicated in Supplementary Tables 7–9. All data for producing Figs. 1–3, and to extract the radial velocity listed in Supplementary Tables 4–6 have been deposited at https://doi.org/10.5281/zenodo.14037031. The authors declare that the data supporting the findings of this study are available in the paper, the supplementary information file, and the Zenodo database. The datasets generated during and/or analyzed during the current study are available from the corresponding author upon request.

## Code availability
The code for generating the SED is publicly available at http://www.hyperion-rt.org/. Stable version 1.4 was used to generate the SED. The evolutionary tracks PARSEC can be found at http://stev.oapd.inaf.it/cgi-bin/cmd (version 3.7). The radial fit was performed with Exo-Stricker, version 0.88, and can be found at https://exo-restart.com/tools/the-exo-striker-tool/. The emcee package is a pure PYTHON package and can be downloaded from https://emcee.readthedocs.io/en/stable/. The ESO pipeline can be downloaded from https://www.eso.org/sci/software/pipelines/.

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

## Acknowledgements

F.P., L.L., and E.B. gratefully acknowledge the Collaborative Research Center 1601 funded by the Deutsche Forschungsgemeinschaft (DFG, German Research Foundation)—SFB 1601 [sub-project A3]—500700252. MZ acknowledges the financial support of the Czech Science Foundation Junior Star grant no. GM24-10599M. VK acknowledges the Czech Science Foundation (ref. 21-06825X).

## Author contributions

F.P. discovered the binary system, performed most of the analysis, and led the writing of the manuscript. M.Z. provided the HR diagram and was responsible for the analysis and calculation of dynamic processes. L.L., A.E., and V.K. contributed to the interpretation of the data. E.B. provided contributions to the background of binaries close to massive stars. M.M. contributed to the SED analysis. M.Z., E.B., M.M., and V.K. improved the text. All authors contributed to the writing of the manuscript.

## Funding

## Competing interests

The authors declare no competing interests.
