## [Transparent Peer Review file · Nature Communications]

A binary system in the S cluster close to the supermassive black hole Sgr A*

Corresponding Author: Dr Florian Peißker

This manuscript has been previously reviewed at another journal. This document only contains reviewer comments and rebuttals for versions considered at Nature Communications. Mentions of the other journal have been redacted.

Version 1:

Reviewer comments:

Reviewer #1

(Remarks to the Author)

The authors claim a discovery of a stellar binary in close proximity to the Galactic center (GC) supermassive black hole (SMBH). As stated previously, if true, the discovery is an interesting addition to the population of known and candidate binaries in the GC: D9 would be the first detection of a binary among the S-stars, the population of stars within approximately 0.04 pc of the supermassive black hole in generally eccentric orbits in a wide range of orbital inclinations around the SMBH. Binaries among these stars are predicted to not be very stable, especially with 3 body interactions with the central black hole, as the authors cite and now more deeply consider. As the candidate binary is a dusty object (a "G object"), a binary object detected among the dusty G-like sources like D9 could also lend strong support for the binary merger formation route for such objects, as the authors now consider more deeply.

The authors have made several improvements to the paper as suggested from the first set of reviews. In particular, the authors have addressed the concerns related to the observational detections: the authors now directly address concerns that a reader may have about the claimed detection by providing additional details about their dataset, data quality, analysis methods, and detection methods. The newly included section discussing the "uniqueness of the IFU data points" now provides necessary details about the authors exploration of false positives and background emission, and the extraction of the RV signal measurements. Furthermore, the inclusion of additional data with ERIS also supports their detection. The authors also clarify how their different analysis procedures are applied to the different parts of the dataset (between the IFU spectroscopic data used to obtain the radial velocity measurements and the imaging data used to obtain photometric flux).

Overall, the changes describing the dataset and analysis procedure have greatly improved the presentation of the results and make it more convincing. The discovery, as presented, now appears to be much more robust.

After these changes, I still have some comments about the current manuscript detailed below, which would be helpful to address.

Abstract:

* Uncertainties should be reported on the binary parameter estimates. Furthermore, since it seems that an edge-on inclination was used to derive these estimates, this assumption should be made clear in the abstract.

* The value in the abstract for the tidal disruption radius is different than what you calculate in the main text (line 293). Which is the correct value?

* You highlight in the main text that binary merger is a much more likely end route for this binary, since it is a dynamically hard system and such systems become harder due to surrounding interactions. If so, I think this should be more conclusively stated in the abstract, especially in the context of the very young age you derive for the system from your SED fits.

Fit of the SED to the YSO model

How is the line of sight extinction taken into account for the SED fit? It does not appear to be referenced in the text how this was accounted for. Extinction is very variable at the Galactic center at small scales (e.g., Schodel+ 2010), so it would be an important effect to consider. For example, in figure 10, differing extinction values and different extinction laws can easily

tweak what SEDs best fit your photometric observations. How would assumptions of different extinction amounts or using different extinction laws towards the D9 source affect the parameters derived? Is there a systematic bias in the derived parameters from the extinction?

LOS velocity offset for RV model

You've derived a Keplerian orbit for D9; can that Keplerian model's RV predictions be used as the LOS RV offset for D9? This would be more accurate than the constant RV offset used in the RV model fit (shown in Figure 13). From your orbit fit, how much is that RV expected to change over the time baseline?

Line 220:

The disk mass found has been stated with a high precision. What are the uncertainties on this value? I would expect that there are uncertainties from the flux estimates that would propagate to the disk mass estimate.

Line 228:

What are the uncertainties on the age estimate? It seems that there likely would be significant uncertainties on this age originating from, for example, the fitting of the SED, biases from the extinction law, constraints on the luminosity from photometry?

Line 304 and onwards:

You conclude that the system is "prone to evaporation" (line 316), but you say that the binary is also dynamically hard earlier on the page (line 296). These two statements seem inconsistent. Only dynamically soft binaries are prone to becoming unbound and subsequent evaporation due to interactions with field stars (e.g., Alexander & Pfuhl, 2014 and Heggie 1975); whereas dynamically hard stars tighten and eventually merge. Can you please clarify what you mean here?

How are the binary evaporation constraints affected if a non-edge on orientation of the binary is assumed?

Figure 5:

It is hard to visually read the histogram with the chosen color schemes and overlapping shapes. Therefore it is difficult to evaluate from this figure the robustness of the image sharpener.

Image sharpener

Is the additional 2–3% deviation in flux density from the image sharpening (described in line 72) applied as an additional source of uncertainty on your derived fluxes?

Minor comments

In paragraph 1, you mention that all bright and massive stars in the S cluster are young stars with ages 4–6 Myr. However there are several identified late-type giants with ages of >3 Gyr (see for example, Habibi+ 2019: Spectroscopic Detection of a Cusp of Late-type Stars around the Central Black Hole in the Milky Way). The statement should be clarified to include this important caveat.

Par. 2, line 87: You start discussing the NIRC2 imagery section in the middle of this paragraph. As you've clarified in the updated manuscript and stated in the rebuttal, the treatment of the IFU data and the imaging photometric data are very different. It would be helpful to divide your discussion about the IFU, spectroscopic data analysis from the imagery data analysis in the text, such as by separating these different data into individual paragraphs. This would help clarify any possible confusion by the reader which treatment is applied to which portion of your dataset.

Figure 1: your text has been discussing the imaging data with filter names (like H, K, or L) but your figure labels each panel as wavelength. To assist the reader, the figure should include both filter names and wavelengths to more easily allow referencing the figure while reading the text.

Reviewer #2

(Remarks to the Author)

This is the second report on the manuscript "The first detected binary system in the S cluster close to the supermassive black hole Sgr A*" by Peisker et al. (the first was when this paper was submitted to [REDACTED]).

While the manuscript has improved, I still have concerns. In my opinion, the results are still overstated in terms of their importance. Additionally, the section where the authors discuss an estimate of the cluster's relaxation timescale based on the age of the binary, and its association with hyper-velocity stars, is problematic. Both arguments have questionable validity. What the authors calculate is not the two-body relaxation time of the central cluster but rather the relaxation time at the distance of the orbiting

binary. One might question to what extent this provides insight into the dynamical processes in the Galactic center.

The authors' response to my earlier report is disappointing. Several arguments they present in their paper are physically incorrect, yet they attempt to refute my concerns. Some of these arguments pertain to fundamental physical gravitational processes, which cannot be dismissed.

Line 307: The authors suggest "that the system is prone to evaporation." It is unclear what this means. If they are implying that the system will become wider, resulting in the destruction of the binary, a thorough dynamical analysis, such as integrating the equations of motion of the triple system over a secular timescale, is necessary. Without this analysis, it is unclear what will happen to the binary, as it depends on other (unknown) orbital parameters and environmental effects. The binary would still be destroyed (merge or disrupted), but the constituent stars would be ejected violently: the authors' suggestion that the destruction of the binary leads to a hyper-velocity star is incorrect.

The manuscript still continues to mix and match units and terms, leading to confusion. For example, in the sentence on lines 305 and 306, 10^6 years and Myr are mixed within one sentence.

The remark on line 187+ is confusing: "Furthermore, the binary system is obviously shielded against the gravitational imprint of Sgr A* since it forms a stable setup that can be observed over almost 20 years." In addition, I see no basis for the term "obviously" here.

The paper continues to make suggestions and implications mixed together non-transparently. This happens near line 222, where the authors make the bold unsupported claim that "Possibly, this ongoing interaction, including the stellar winds of the S cluster, will disperse the disk of D9 in the future [36, 37]." and later near line 323, the authors try to connect their result to the possibility that all the S-stars were binaries once. I do not think that we know anything of the binary nature of the S-stars, except for the few cases discussed in this manuscript. Extrapolating the result to all S-stars seems excessive. I would hope that the authors stay to facts, and leave the bold extrapolations to thorough future analysis.

Version 2:

Reviewer comments:

Reviewer #1

(Remarks to the Author)

The authors claim a discovery of a stellar binary in close proximity to the Galactic center (GC) supermassive black hole (SMBH). As stated previously, if true, the discovery is an interesting addition to the population of known and candidate binaries in the GC and an exciting source of support for the binary merger hypothesis that has been proposed to be a frequent occurrence near the GC SMBH. D9 would be the first detection of a binary among the S-stars, the population of stars within approximately 0.04 pc of the supermassive black hole in generally eccentric orbits in a wide range of orbital inclinations around the SMBH. Binaries among these stars are predicted to not be very stable, especially with 3 body interactions with the central black hole, as the authors cite and now explore in sufficient detail. As the candidate binary is a dusty object (a "G object"), a binary object detected among the dusty G-like sources like D9 lends strong support for the binary merger formation route for such objects, especially since the authors now describe in detail that a merger is likely the end route for their discovered binary system.

The updates to the submitted paper have greatly improved the presentation of the authors' results and address the major comments that I previously submitted about their work. The authors now consider additional sources of uncertainties that were highlighted in previous rounds of review, and uncertainties are now more clearly reported for their main results about their discovery. Additionally, the few discrepancies across the main text, abstract, and methods that I had previously highlighted about the results have been amended. I have no more major comments for the submitted paper. I have some minor comments about the revised manuscript which I detail further down in my comments.

Notably, the authors' findings lend strong credibility to the binary merger hypothesis that has been proposed as the origin of

the G objects. The text and abstract now detail this implication in the necessary detail. The authors have also fixed the previous contradiction in the text about the system being prone to “binary evaporation”. The newly added section titled “Dynamical processes and stellar populations” greatly assists in providing evidence to support their analysis and claims about the dynamical nature of the D9 object. The text now appropriately highlights that a binary merger is a likely near-future outcome for this particular object.

The authors now also explore the consequences of line of sight extinction when determining physical parameters from their photometric measurements. Since extinction is large and variable on small angular scales towards the GC, this is an important consideration. In the new section titled Extinction, the authors now describe the robustness of their results against these considerations and incorporate the resulting uncertainties in their derivation of the main results.

Minor comments

Line 355: It's important to note that the binary fraction constraint numbers being quoted from the cited experiments are actually lower and upper limits at 1 sigma, respectively, rather than precise measurements. So instead of just quoting the numbers as 72% and 17% as it is currently in the text, it would be more accurate to be reported as $\geq 72\%$ and $\leq 17\%$, at 68% confidence.

Figure 11: The point and label for your newly discovered binary D9 are difficult to read in this plot. Consider a different color or different emphasis for D9 in this figure to more clearly highlight the significant implication about D9's likely future merger.

Table 6: The column name “Period” can be confusing in the context of a paper about a binary star since Period would be used to imply the orbital period of the system. Perhaps consider using the name “Observation Epoch” or “Observation Time” to describe the time of your observations.

Reviewer #2

(Remarks to the Author)

Here are my comments to the manuscript "The first detected binary system in the S cluster close to the supermassive black hole Sgr A*" by Peisker et al. I have to say the the paper improved considerably, and only a few concerns remain.

the term τ_{Kozai} should be called τ_{vZLK} , or τ_{LK} , and please, define the term. The term $\tau_{\text{Kozai}}^{\text{SMBH}}$ is even more confusing. The von Zeipel-Lidov-Kozai process operates on three particles that orbit each other hierarchically. It is quite uncommon to specifically refer to any of the individual bodies as being the cause of it; although one could argue that the outer-most body (here the SMBH) Lidov-Kozai's the binary D9. On conferences, one sometimes uses such slang, but I would refrain from such uncommon terminology in a publication. Also, you never defined the term. BTW. there is no confusion on which star is affecting which binary.

Referring to the orbital parameters of the binary D9 as "Secondary Keplerian Parameters" is confusing at best. Why not just keep referring to the orbital parameters of D9, and those of D9 in orbit around Sgr A*. Then nobody can be confused about this. Same with referring to e_{sec} and a_{sec} , etc. Better to write e_{D9} and a_{D9} , etc. The indices for the orbital elements for the binary D9 around Sgr A* can then be omitted.

More general, and surely not confusing might be to refer to an "inner" binary (a_{in} and e_{in}), and an "outer" binary (a_{out} and e_{out}).

Regarding some more detailed comments:

Terminology is still somewhat confusing in places. For example, Table 1 refers to M_{D9} , but lists the mass of the primary star, rather than the total binary mass. Later, in the text, this mass is referred to as $M_{\text{D9, prim}}$. Maybe, the authors can refer to m_{D9a} , and m_{D9b} ?

line 76: remove "essential"

line 313: It is not relevant, and confusing, to refer to D9 in orbit

around Sgr A* as a hard binary. It is rather obvious, as its orbit is completely dominated by the black hole (as also, the binary is very much inside the sphere of influence of the black hole). It is much more interesting to discuss the binary D9, which is soft.

line 318: remove "so called" line 319: please call it "von Zeipel Lidov Kozai" or "Lidov Kozai". remove reference #4, and add a reference to von Zeipel. It is better to list the author's names in chronological order. Including the reference #4 is inappropriate, and seems to suggest that they contributed to the fundamental process, which they did not. von Zeipel, however, definitely did.

line 324: what is "degenerate" about this? line 324: why use the terms "interference on the cluster members". For both terms "degenerate" and "interference" it is unclear what the authors mean, or try to say. Please, stick to the common scientific terminology.

line 339: it is unclear what is meant with "destabilization between the primary and secondary". Maybe, the authors try to indicate that the binary becomes unbound or enters a democratic resonance?

line 341: please replace "could be a product of" with "has a contribution from".

line 343: how is the destruction rate of 1 Myr for D9 calculated?

line 344: I have some problems with the term "likely", as it is unspecific and should be associated with a probability. Maybe the authors can consider using "may have". It's a bit weaker term, maybe, but probably more appropriate in this context.

line 347: please: "Kozai-Lidov relaxation timescale" is not standard, and I encourage the authors to change it. In fact, I do not know what the authors try to say here (see also line 364). Maybe, the authors mean the "Lidov-Kozai period"?

line 349: remove "majority" line 350: replace "must" for a more appropriate term. line 346: remove "fully".

line 367-370: please remove altogether. I see no point in referring to future observations, or upcoming surveys or instruments. They will always improve the data; it would be something if the data got worse as a result of new instruments.

I did go through the Methods section rather quickly. Had read it before, and it is mostly observational, some statistical analysis, and MCMC. My only comment is that the right-most column of table 2 is unspecific, and gives inappropriate trust in the results by naming it the "uncertainty". This, clearly, is not the right term for the geometric difference between the Best-fit value and the MCMC results. The authors should include the MCMC 90% confidence interval or the MCMC standard deviation from the combined posteriors.

Dear referee #1,

Thank you very much for the helpful comments. We think they greatly improved the paper and especially the discussion. From our point of view, it seems plausible that the idea of binary mergers is more likely with the detection of D9. Again, thanks for pointing this out. Please find our response to your last report below. All the major changes are listed separately. All the changes are highlighted in bold in the manuscript.

#####

Major changes:

- Added a new subsection, "Extinction"
- Added all the intermediate results to calculate the final plotted velocity (Table 9, 10, 11)
- Added a new subsection, "Age of the D9 binary system"
- Added a new subsection, "Dynamical processes and stellar populations"
- Moved the HR diagram to this subsection
- Updated Figure 10
- Added Figure 11

Minor changes:

- Typos
- Reformulated the text for clarity whenever feasible
- Fig. 8/12-> changed v_{mod} to v_{norm}
- Updated the abstract (structure/content)
- Updated the abstract (weakened statements about the binary and S star connection)
- Updated the main text and Methods section based on the comments
- Replace Myr with 10^6 yr
- Deleted the two-body relaxation timescale
- Included Kozai-Lidov timescales as the mechanism that might drive the binary to merge

#####

Abstract:

Uncertainties should be reported on the binary parameter estimates. Furthermore, since it seems that an edge-on inclination was used to derive these estimates, this assumption should be made clear in the abstract.

We included the uncertainties and made it clear that the inclination is an assumed quantity.

The value in the abstract for the tidal disruption radius is different than what you calculate in the main text (line 293). Which is the correct value?

Yes, we noticed this inconsistency and corrected it. Thanks for pointing this out!

You highlight in the main text that binary merger is a much more likely end route for this binary, since it is a dynamically hard system and such systems become harder due to surrounding interactions. If so, I think this should be more conclusively stated in the abstract, especially in the context of the very young age you derive for the system from your SED fits.

We updated the abstract based on your suggestion and stressed the fate of the binary as a merger.

Fit of the SED to the YSO model

How is the line of sight extinction taken into account for the SED fit? It does not appear to be referenced in the text how this was accounted for. Extinction is very variable at the Galactic center at small scales (e.g., Schoedel+ 2010), so it would be an important effect to consider. For example, in figure 10, differing extinction values and different extinction laws can easily tweak what SEDs best fit your photometric observations. How would assumptions of different extinction amounts or using different extinction laws towards the D9 source affect the parameters derived? Is there a systematic bias in the derived parameters from the extinction?

Yes, this is correct. But all the flux-density values are extinction-corrected. Since we used the most observed source of the Galactic center (besides Sgr A*) as a reference, i.e. S2/S0-2, we are confident about the extinction and the reference magnitudes.

We indicated this in the text.

Furthermore, we tested different extinction values for D9. For this, we implemented a new subsection to investigate the impact further. In this subsection, we conclude that we find no significant impact on the derived stellar parameters and that the analysis is robust against variations of the extinction.

LOS velocity offset for RV model

You've derived a Keplerian orbit for D9; can that Keplerian model's RV predictions be used as the LOS RV offset for D9? This would be more accurate than the constant RV offset used in the RV model fit (shown in Figure 13). From your orbit fit, how much is that RV expected to change over the time baseline?

As it is mentioned in the text, the LOS RV offset for D9 is assumed to be constant inside the uncertainties. Since D9 is on the descending part of the orbit, the intrinsic LOS velocity and its change over the investigated epochs are smaller than the uncertainties for the periodic signals (see Fig. 3 which displays the low proper motion of D9). We estimated a change of the LOS of the intrinsic velocity of ± 15 km/s which is compatible with the ± 17 km/s from the individual measurements of the periodic signal. Of course, forthcoming observations should take this into account. We indicated this in the main text.

Line 220:

The disk mass found has been stated with a high precision. What are the uncertainties on this value? I would expect that there are uncertainties from the flux estimates that would propagate to the disk mass estimate.

Yes, you are right. We have updated Table 1 and the text. The construction of the SED by the SEDfitter incorporates the flux density uncertainties. Therefore, all the uncertainties for the parameters of D9 are based on the flux density uncertainties and the comparison with 10000 different individual models.

Line 228:

What are the uncertainties on the age estimate? It seems that there likely would be significant uncertainties on this age originating from, for example, the fitting of the SED, biases from the extinction law, constraints on the luminosity from photometry?

Based on the extinction, we derived different results for the temperature and radius of the system. These values are covered by the initial uncertainties. However, we use these “new” values to update the HR diagram and indicate a range for the plausible age of the system as based on the range of stellar masses (2.4-2.8 Msun) consistent with the position of D9 on the HR diagram. We updated the text and implemented the uncertainties for the age.

Line 304 and onwards:

You conclude that the system is “prone to evaporation” (line 316), but you say that the binary is also dynamically hard earlier on the page (line 296). These two statements seem inconsistent. Only dynamically soft binaries are prone to becoming unbound and subsequent evaporation due to interactions with field stars (e.g., Alexander & Pfuhl, 2014 and Heggie 1975); whereas dynamically hard stars tighten and eventually merge. Can you please clarify what you mean here?

Yes, you are correct, this was also pointed out by referee #2. This was a misleading and confusing description of the system. We have corrected it. The binary is hard, we expect it to merge in the future. We adjusted the text accordingly.

How are the binary evaporation constraints affected if a non-edge on orientation of the binary is assumed?

The edge-on orientation in the dense S cluster is only relevant for the observer and should not affect any dynamics regarding the binary. We added a sentence in the discussion part to stress this.

Figure 5:

It is hard to visually read the histogram with the chosen color schemes and overlapping shapes. Therefore it is difficult to evaluate from this figure the robustness of the image sharpener.

Yes, we updated the representation. However, we would like to emphasize that all the estimated fluxes and median values are mentioned in the text and listed in Table 3.

Image sharpener

Is the additional 2–3% deviation in flux density from the image sharpening (described in line 72) applied as an additional source of uncertainty on your derived fluxes?

For the uncertainties of the flux density, we are in favor of a conservative approach. The chosen uncertainties are in the range of ~15-30% which underlines this. The 2-3% are already included.

Minor comments

In paragraph 1, you mention that all bright and massive stars in the S cluster are young stars with ages 4–6 Myr. However there are several identified late-type giants with ages of >3 Gyr (see for example, Habibi+ 2019: Spectroscopic Detection of a Cusp of Late-type Stars around the Central Black Hole in the Milky Way). The statement should be clarified to include this important caveat.

Yes, you are right. We have rephrased this part to make it clear. Furthermore, we added Habibi+2019 as an additional reference wherever appropriate.

Par. 2, line 87: You start discussing the NIRC2 imagery section in the middle of this paragraph. As you've clarified in the updated manuscript and stated in the rebuttal, the treatment of the IFU data and the imaging photometric data are very different. It would be helpful to divide your discussion about the IFU, spectroscopic data analysis from the imagery data analysis in the text, such as by separating these different data into individual paragraphs. This would help clarify any possible confusion by the reader which treatment is applied to which portion of your dataset.

This is a good point. We separated both data types into individual paragraphs.

Figure 1: your text has been discussing the imaging data with filter names (like H, K, or L) but your figure labels each panel as wavelength. To assist the reader, the figure should include both filter names and wavelengths to more easily allow referencing the figure while reading the text.

This is also a good point, we corrected it.

Dear referee #2,

Thank you very much for your comments. As you will see, we have addressed all the raised concerns and implemented additional information to support our claims. We would like to bring your attention to the new subsection, "Dynamical processes and stellar populations" where we discuss the possible fate of the binary system. In this subsection, we analyze merging and disruption scenarios and find that the most likely evolutionary path for the D9 binary system can be described as a merger which strongly supports the idea that the G population might harbor mergers, as was first proposed by Ciurlo et al. (2020).

In the following, we list all the major and minor changes made in the manuscript.

#####

Major changes:

- Added a new subsection, "Extinction"
- Added all the intermediate results to calculate the final plotted velocity (Table 9, 10, 11)
- Added a new subsection, "Age of the D9 binary system"
- Added a new subsection, "Dynamical processes and stellar populations"
- Moved the HR diagram to this subsection
- Updated Figure 10
- Added Figure 11

Minor changes:

- Typos
- Reformulated the text for clarity whenever feasible
- Fig. 8/12-> changed v_{mod} to v_{norm}
- Updated the abstract (structure/content)
- Updated the abstract (weakened statements about the binary and S star connection)
- Updated the main text and Methods section based on the comments
- Replace Myr with 10^6 yr
- Deleted the two-body relaxation timescale
- Included Kozai-Lidov timescales as the mechanism that might drive the binary to merge

#####

While the manuscript has improved, I still have concerns. In my opinion, the results are still overstated in terms of their importance. Additionally, the section where the authors discuss an estimate of the cluster's relaxation timescale based on the age of the binary, and its association with hyper-velocity stars, is problematic. Both arguments have questionable validity.

1. Yes, indeed. The terms and nomenclature used were misleading: the cluster relaxation timescales estimated by Morris (1993) are not related to the S cluster nor to D9. They have been used to explain the migration timescales for stars outside of the Nuclear Stellar Cluster into the inner parsec. The relaxation timescale calculated in the text is the stellar relaxation timescale, as you correctly pointed out: **What the authors calculate is not the two-body relaxation time of the central cluster but rather the relaxation time at the distance of the orbiting binary.** -> The theoretical framework for this calculation was formulated by Alexander & Pfuhl (2014). The authors described the relaxation timescale as “stellar relaxation timescale” in their paper and described the dynamical interaction with the dark cusp. However, we noticed that this description might be misleading and could create confusion caused by the terminology used in the review by Alexander (2005), where the author defined the two-body relaxation timescale, which is in the range of 10^9 years. We have exchanged this with the KL timescales ($\sim 10^6$) and recalculated the stellar relaxation timescale. We noticed an error in our calculations and derived $\sim 10^9$ years (by coincidence, it approximately coincides with the two body relaxation timescale for massive stars outside the inner parsec). Hence, the dark cusp inside the S cluster does not have a significant impact on the evolution of the binary. We included all estimates in Fig. 11, and rephrased the abstract and the main text at the related position.
2. We agree that the association with HV stars is misleading and confusing. We have eliminated the connection to HV stars.

In my opinion, the results are still overstated in terms of their importance.

Yes, we have weakened most of our claims and rephrased the abstract and the text accordingly. As explained in the next comment, we still believe the results are important and worth publishing since the binary on the scale of ~ 40 mpc from Sgr A* is relevant for estimating the binary fraction in the central parts of the nuclear star cluster.

One might question to what extent this provides insight into the dynamical processes in the Galactic center.

The Galactic center is a highly dynamical environment with different large- (Clockwise Disk, IRS 13, etc) or small-scale (S cluster, Sgr A*) structures. All of these regions require a different set of approaches and tools. At no point do we want to extrapolate our findings to the entire Galactic center. Please feel free to indicate the location of the manuscript where we do so.

But since there have been various attempts to explain the unusual low binary rate of the S cluster, we think that the identification of a binary is relevant.

However, we used your comments to expand our discussion about the dynamical processes that might be relevant for the evolution of D9. We have added a new subsection “Dynamical processes and stellar populations” in the methods Section.

The authors' response to my earlier report is disappointing.

We are sorry to read this. We have re-evaluated your first report and isolated points that should have been addressed in the last submitted revision. For example, we moved the HR diagram to a designated subsection in the Methods section. Since the representation of the Figure itself is in agreement with similar studies in the literature, we think that adjusting it leaves out important information that are relevant for the reader. Since D9 is a part of the G object population, we implemented G2, X3, and X7 as they were recently covered in the literature and might be important for comparison.

In addition, we weakened some of our claims and rephrased the text/abstract.

Furthermore, we included the subsection “Dynamical processes and stellar populations” to discuss different dynamical processes for D9. We discuss merging and disruption events and find that the system might evolve into a merger.

Line 307: The authors suggest "that the system is prone to evaporation." It is unclear what this means. If they are implying that the system will become wider, resulting in the destruction of the binary, a thorough dynamical analysis, such as integrating the equations of motion of the triple system over a secular timescale, is necessary. Without this analysis, it is unclear what will happen to the binary, as it depends on other (unknown) orbital parameters and environmental effects. The binary would still be destroyed (merge or disrupted), but the constituent stars would be ejected violently: the authors' suggestion that the destruction of the binary leads to a hyper-velocity star is incorrect.

Indeed, this was an incorrect mix of arguments, thank you for pointing this out. We rephrased this part and emphasized that the system might merge. There is no explicit connection to HV stars anymore. We specifically exclude this connection.

The manuscript still continues to mix and match units and terms, leading to confusion. For example, in the sentence on lines 305 and 306, 10^6 years and Myr are mixed within one sentence.

Thanks for pointing this out. We have corrected this.

The paper continues to make suggestions and implications mixed together non-transparently. This happens near line 222, where the authors make the bold unsupported claim that "Possibly, this ongoing interaction, including the stellar winds of the S cluster, will disperse the disk of D9 in the future [36, 37]."...

In the Orion Nebula, it has been observed that the presence of nearby stars results in the photoevaporation of disks. The presence of a companion can also result in the decreased lifetime of the circumprimary disk. We have slightly rephrased the sentence and updated the references.

...and later near line

323, the authors try to connect their result to the possibility that all the S-stars were binaries once. I do not think that we know anything of the binary nature of the S-stars, except for the few cases discussed in this manuscript. Extrapolating the result to all S-stars seems excessive. I would hope that the authors stay to facts, and leave the bold extrapolations to thorough future analysis.

We are sorry for the confusion. But we are not saying that all the S stars were binaries once. However, there is compelling evidence that O and B stars exhibit a binary fraction of almost 100% (please see Offner+2023). The question should be raised as to why there are no binaries for the bright and massive S stars (which are classified spectrally as B-type stars, please see Habibi+2017). We specifically mention this: "...young stars in the immediate vicinity of Sgr A* lost their putative...". But you are right, Habibi+2019 identified a population of old stars that might have migrated into the S cluster. The history of the putative companions of these stars is unclear, which is why the statement "all S stars were once binaries" is incorrect.

We rephrased this specific part of the discussion, and weakened our corresponding statements, and limited our conclusions to the B-type stars.

Dear referee #1,

Thanks a lot for your final comments and all the previous ones. The manuscript strongly improved not only on the wording but also regarding its content and interpretation. Please see our comments below.

#####

Line 355: It's important to note that the binary fraction constraint numbers being quoted from the cited experiments are actually lower and upper limits at 1 sigma, respectively, rather than precise measurements. So instead of just quoting the numbers as 72% and 17% as it is currently in the text, it would be more accurate to be reported as $\geq 72\%$ and $\leq 17\%$, at 68% confidence.

Thanks for pointing this out. We added the relevant information in the text.

Figure 11: The point and label for your newly discovered binary D9 are difficult to read in this plot. Consider a different color or different emphasis for D9 in this figure to more clearly highlight the significant implication about D9's likely future merger.

We changed the color of the D9 point to red with the white face and we made the D9 label bold, hence we hope that the point is more noticeable now. We also changed the abbreviation of the Lidov-Kozai mechanism to vZLK (von Zeipel-Lidov-Kozai mechanism) as requested by the second referee.

Table 6: The column name "Period" can be confusing in the context of a paper about a binary star since Period would be used to imply the orbital period of the system. Perhaps consider using the name "Observation Epoch" or "Observation Time" to describe the time of your observations.

Good point, we changed it to Epoch. Alternatively, we could use the name "Date". However, in the context of observations, a common term used is "Epoch".

#####

#review 2

Dear referee #2,

Thanks a lot for your comments and suggestions. Especially regarding the dynamical timescales and the discussion about the vZLK process helped to improve the manuscript and its interpretation enormously. Please see our comments below.

#####

the term τ_{Kozai} should be called τ_{vZLK} , or τ_{LK} , and please, define the term. The term $\tau_{\text{Kozai}}^{\text{SMBH}}$ is even more confusing. The von Zeipel-Lidov-Kozai process operates on three particles that orbit each other hierarchically. It is quite uncommon to specifically refer to any of the individual bodies as being the cause of it; although one could argue that the outer-most body (here the SMBH) is the cause of the Lidov-Kozai's the binary D9. On conferences, one sometimes uses such slang, but I would refrain from such uncommon terminology in a publication. Also, you never defined the term. BTW. there is no confusion on which star is affecting which binary.

In the Sections "Dynamical processes and stellar populations" and "Fate of the binary" we changed Kozai-Lidov to vZLK as suggested by the referee. We reformulated the sentences concerning the vZLK mechanism, focusing on the hierarchical setup of the system involving the D9 binary, avoiding the words such as "driven by".

Referring to the orbital parameters of the binary D9 as "Secondary Keplerian Parameters" is confusing at best. Why not just keep referring to the orbital parameters of D9, and those of D9 in orbit around Sgr A*. Then nobody can be confused about this. Same with referring to e_{sec} and a_{sec} , etc. Better to write e_{D9} and a_{D9} , etc. The indices for the orbital elements for the binary D9 around Sgr A* can then be omitted.

More general, and surely not confusing might be to refer to an "inner" binary (a_{in} and e_{in}), and an "outer" binary (a_{out} and e_{out}).

Terminology is still somewhat confusing in places. For example, Table 1 refers to M_{D9} , but lists the mass of the primary star, rather than the total binary mass. Later, in the text, this mass is referred to as $M_{\text{D9, prim}}$. Maybe, the authors can refer to m_{D9a} , and m_{D9b} ?

We combined all three comments to address them simultaneously. First of all, binary systems are described in general as the primary and the secondary. We agree that some readers not familiar with this classification might get confused about the terms. Hence, we included a short sentence describing this and, wherever applicable, changed the indices. You are right, readers might get confused about the mass in Table 1. For consistency, we used the term $M_{\text{D9, prim}}$ and transferred these changes to the main text.

line 76: remove "essential"

Yes, essential might be a little bit exaggerated. We exchanged it with "important."

line 313: It is not relevant, and confusing, to refer to D9 in orbit around Sgr A* as a hard binary. It is rather obvious, as it's orbit is completely dominated by the black hole (as also, the binary is very much inside the sphere of influence of the black hole). It is much more interesting to discuss the binary D9, which is soft.

Although it might be obvious for some readers, we included this discussion to provide an inclusive discussion for an audience that might not be familiar with the topic. With the tidal (Hill) radius, the obvious information becomes accessible to readers who might be wondering about the presence of the binary close to the supermassive black hole.

line 318: remove "so called" line 319: please call it "von Zeipel Lidov Kozai" or "Lidov Kozai". remove reference #4, and add a reference to von Zeipel. It is better to list the author's names in chronological order. Including the reference #4 is inappropriate, and seems to suggest that they contributed to the fundamental process, which they did not. von Zeipel, however, definitely did.

At the beginning of the Section "Dynamical processes and stellar populations", we introduced von Zeipel-Lidov-Kozai mechanism in the chronological order and we also referred to all of the three relevant papers. We included the reference to Stephan et al. (2016) at a more appropriate position. The von Zeipel Lidov Kozai (vZLK) is now cited in the correct order everywhere.

line 324: what is "degenerate" about this? line 324: why use the terms "interference on the cluster members". For both terms "degenerate" and "interference" it is unclear what the authors mean, or try to say. Please, stick to the common scientific terminology.

We excluded both terms and slightly reformulated the text.

line 339: it is unclear what is meant with "destabilization between the primary and secondary". Maybe, the authors try to indicate that the binary becomes unbound or enters a democratic resonance?

Yes, you are right, destabilization is a confusing term and is not correct in this context. We slightly reformulated the text.

line 341: please replace "could be a product of" with "has a contribution from".

Very good point. We corrected it.

line 343: how is the destruction rate of 1Myr for D9 calculated?

We replaced destruction with the Lidov-Kozai timescale, which was calculated before. We are sorry for the confusion.

line 344: I have some problems with the term "likely", as it is unspecific and should be associated with a probability. Maybe the authors can consider using "may have". It's a bit weaker term, maybe, but probably more appropriate in this context.

We agree, we included "could have" because a few words afterwards, we use the term "may" again.

line 347: please: "Kozai-Lidov relaxation timescale" is not standard, and I encourage the authors to change it. In fact, I do not know what the authors try to say here (see also line 364). Maybe, the authors mean the "Lidov-Kozai period"?

We reformulated this part in the Section "Fate of the binary", and instead refer to the vZLK timescale.

line 349: remove "majority" line 350: replace "must" for a more appropriate term. line 346: remove "fully".

We removed "majority" and "fully" and replaced "must" with "may".

line 367-370: please remove altogether. I see no point in referring to future observations, or upcoming surveys or instruments. They will always improve the data; it would be something if the data got worse as a result of new instruments.

We removed it.

I did go through the Methods section rather quickly. Had read it before, and it is mostly observational, some statistical analysis, and MCMC. My only comment is that the right-most column of table 2 is unspecific, and gives inappropriate trust in the results by naming it the "uncertainty". This, clearly, is not the right term for the geometric difference between the Best-fit value and the MCMC results. The authors should include the MCMC 90% confidence interval or the MCMC standard deviation from the combined posteriors.

We corrected the term and caption accordingly.